# DEMYSTIFYING APPROXIMATE RL WITH $\epsilon$-GREEDY EXPLORATION: A DIFFERENTIAL INCLUSION VIEW

## ABSTRACT

Q-learning and SARSA(0) with $\epsilon$-greedy exploration are leading reinforcement learning methods, and their tabular forms converge to the optimal Q-function under reasonable conditions. However, with function approximation, these methods exhibit strange behaviors, e.g., policy oscillation and chattering, convergence to different attractors (possibly even the worst policy) on different runs, etc., apart from the usual instability. Accordingly, a theory to explain these phenomena has been a long-standing open problem, even for basic linear function approximation (Sutton, 1999). Our work uses differential inclusion theory to provide the first framework for resolving this problem. We further illustrate via numerical examples how this framework helps explain these algorithms' asymptotic behaviors.

## 1 INTRODUCTION

Tabular versions of value-based Reinforcement Learning (RL) algorithms such as Q-learning and SARSA are known to converge to the optimal Q-function under reasonable conditions (Singh et al., 2000; Jaakkola et al., 1993; Tsitsiklis, 1994). However, the story of their approximate variants (those using *function approximation*) with *$\epsilon$-greedy exploration* has been inconclusive. On the one hand, these variants, e.g., the Deep Q-Network (DQN) (Mnih et al., 2015), have shown significant empirical successes. On the other, there is also growing evidence of undesirable behaviors such as policy oscillation, i.e., indefinitely cycling between multiple policies, or convergence to a sub-optimal or even the worst possible policy (Gordon, 1996; 2000; De Farias & Van Roy, 2000; Bertsekas, 2011; Young & Sutton, 2020). Accordingly, a mathematical framework to explain such behaviors and, in turn, to identify conditions for some minimal reliability has been a long-standing open problem, even in the basic linear function approximation case (Sutton, 1999, Problem 1).

By reliability of an approximate value-based method, we mean some basic notions like i.) stability, ii.) convergence to the optimal policy when the optimal Q-value function lies in the approximating function class, or iii.) convergence to a policy with a better $Q$-value function than the initial policy. Tabular variants of Q-learning and SARSA are reliable in all these three viewpoints under the reasonable conditions that guarantee their convergence (see references above). Likewise, there are sufficient, albeit restrictive conditions (Melo et al., 2008; Chen et al., 2019; Carvalho et al., 2020; Lee & He, 2020; Xu & Gu, 2020) under which approximate Q-learning and SARSA with a fixed behavior policy are reliable, at least as per the first two notions above. For example, these conditions hold when the behavior policy is close to the optimal policy (the one to be estimated).

In contrast, we claim that the reliability of approximate value-based RL methods with $\epsilon$-greedy exploration cannot be taken for granted. To see this, consider Figure 1, showing trajectories of different runs of a variant of DQN. This variant also employs experience replay and a target network as in (Mnih et al., 2015), but uses a *linear* function instead of a neural network for approximating the optimal Q-value function. The reduction in the approximation power is offset by including the optimal Q-value function in this linear function class[1]. For all the trajectories, the starting conditions are the same and set so that the initial behavior policy is close to the optimal policy, aligned with the conditions in the fixed behavior policy literature. Thus, one would expect the greedy policy along these trajectories to converge to the optimal policy. Surprisingly, we observe three different behaviors: i.) convergence to a sub-optimal policy (red), ii.) oscillation between two sub-optimal policies (blue, tail end) and iii.) convergence to the optimal policy (green).

---

[1]This is ensured by setting one column of the state-action feature matrix to the optimal Q-value function.

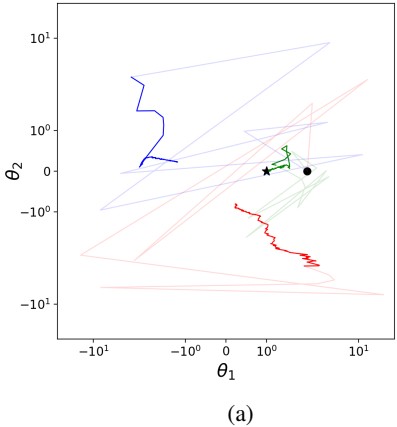
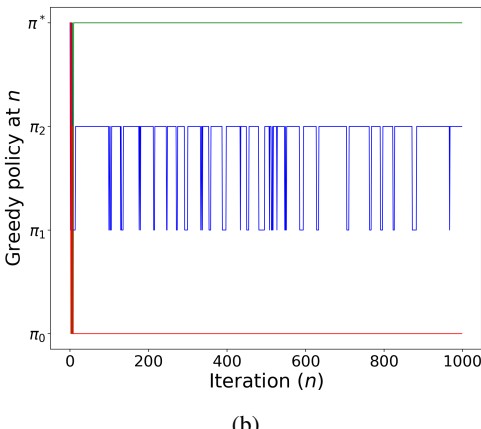

(a)                                                              (b)

Figure 1: Trajectories of three runs of DQN on a 2-state 2-action MDP, with a linear 2-dimensional Q-value approximation which *perfectly represents* the optimal Q-value function $Q^*$. Figure 1a shows these trajectories in the parameter space. The parameters of $Q^*$ are denoted by the black star at $(1, 0)$. The initial parameter for all trajectories is the same (the black dot) and is chosen so that the initial behavior is the $\epsilon$-greedy version of the optimal policy. In this idealized setting, one would have expected all trajectories to go to the star. In reality, all of them do converge, but only the green one has the desired limit (the initial fading of colors is for ease of exposition). Figure 1b shows the greedy policies associated with the different trajectories. The limiting greedy policy for the green trajectory is unique and is the optimal one; for the red, it is some sub-optimal policy. In contrast, that of the blue trajectory *oscillates* between two other sub-optimal policies.

The above discussion along with those in (Young & Sutton, 2020) show that approximate value-based methods with $\epsilon$-greedy exploration can exhibit several pathological behaviors beyond the textbook instability phenomenon (Sutton & Barto, 2018), raising serious doubts about their reliability in practice. Towards addressing these concerns, a theory to explain the limiting behaviors of approximate value-based RL with greedification is thus an extremely important first step.

Existing analyses based on Ordinary Differential Equations (ODEs) are of limited utility in building such a theory. To see why, note that RL schemes can be viewed as update rules of the form

$$\theta_{n+1} = \theta_n + \alpha_n[h(\theta_n) + M_{n+1}], \qquad n \geq 0, \tag{1}$$

where $h$ is some driving function, $\alpha_n$ is the stepsize, and $M_{n+1}$ is noise. When $h$ is 'nice' overall, e.g., globally Lipschitz continuous, the ODE method is useful to show that the limiting dynamics of (1) is governed by the ODE $\dot{\theta}(t) = h(\theta(t))$ (Benaïm, 1999; Borkar, 2009). This is indeed the case in policy evaluation. In value-based RL methods, however, $h$ is quite complex, even with linear function approximation: the update rules involve sampling from distributions which change depending on the iterates. Accordingly, the ODE method has been made to work here only via restrictive assumptions on the sampling distribution, e.g., fixed behavior policy (Carvalho et al., 2020), near-optimal behavior policy (Melo et al., 2008; Chen et al., 2019), smooth soft-max behavior policy (Perkins & Precup, 2002; Zou et al., 2019), etc. With $\epsilon$-*greedy exploration*, the situation is even worse since, as we show, the resultant dynamics also turns out to be *discontinuous*.

**Key Contributions**: The main highlights of this work can be summarized as follows.

1. *Analysis Framework*: Our work uses Differential Inclusion (DI) theory (Aubin & Cellina, 2012) to develop a new framework for analyzing value-based RL methods. Our key steps include i.) breaking down the parameter space into regions where the algorithm's dynamics are *simple*, ii.) identifying a DI that *stitches* the local dynamics together, and iii.) using this DI to explain the algorithm's overall (possibly complex) behavior. A DI is a generalization of an ODE which enables this stitching by allowing for *multiple* update directions at every point.

2. *Explanation of Asymptotic Behaviors*: We demonstrate our idea for a variant of linear[2] Q-learning and SARSA(0) with $\epsilon$-greedy exploration. We show that the DIs uncovered by our framework govern the asymptotic behaviors of these algorithms. Our work thus provides the first pathway to systematically explain the whole range of limiting phenomena that approximate value-based RL methods with greedy exploration can exhibit, and answers the question posed by Sutton (1999).

   The Q-learning and SARSA(0) variants we study do not involve experience replay and the target network as in DQN, but instead sample independently[3] from the stationary distribution of the current behavior policy. Our motivation for these variants is threefold: i.) the independent sampling serves as an idealized replay buffer, ii.) their asymptotic behaviors are amenable to detailed analysis, and iii.) they still show all limiting behaviors of the kind we observe in Figure 1.

3. *Illustrations*: In Section 3, we present case studies that isolate and spotlight the different types of behaviors of the kind shown in Figure 1 (unique or multiple attractors, policy oscillations). In each study, we explicitly describe how to identify the appropriate DI using our framework, what the nature of its solutions are, and how they describe the asymptotic behavior of the algorithm. In particular, we explain policy oscillation via a hitherto unseen 'sliding mode' phenomenon.

**Related Work**: Several works identify and report a variety of complex behaviors for approximate algorithms. These include the classic example of instability for linear Q-learning (Baird, 1995) and the phenomenon of chattering in linear SARSA(0) (Gordon, 1996) and its approach to a bounded region (Gordon, 2000). Apart from these, De Farias & Van Roy (2000) show how approximate value iteration may not possess any fixed points, while Bertsekas (2011) argues that approximate policy iteration schemes may generally be prone to policy oscillations, chattering, and convergence to poor solutions. More recently, Young & Sutton (2020) and Schaul et al. (2022) show experimentally how a range of approximate value-based RL methods involving greedification exhibit pathological behaviors such as policy oscillation, multiple fixed points, and consistent convergence to the 'worst' policy. Our framework helps rigorously explain all these phenomena.

On the theoretical front, while global convergence results exist for Q-learning and SARSA in the tabular setting, their analysis with function approximation presents significant challenges. Even with arguably simple linear function approximation, the general form of the iterations is nonlinear, and there is no natural contraction or Lyapunov function structure that can be exploited. One prominent stream of work focuses on extending linear SA theory to analyze Q-learning with linear function approximation (Melo et al., 2008; Carvalho et al., 2020; Chen et al., 2019) and nonlinear (neural) function approximation (Fan et al., 2020; Xu & Gu, 2020). However, all these works *hold the behavior policy fixed*, along with imposing other conditions on it (e.g., being close to the optimal policy). These assumptions effectively ensure that the resulting nonlinear SA has a Lyapunov function and thus convergence guarantees. Another such notable work is that of Lee & He (2020), which uses ideas from switched system theory to analyze nonlinear ODEs. This is perhaps similar in spirit to our endeavor as switched systems are also useful for analyzing discontinuous dynamics.

There is also a set of analyses that apply SA techniques to study SARSA(0) with incrementally changing policies (Perkins & Precup, 2002; Melo et al., 2008; Zhang et al., 2022; Zou et al., 2019). However, these apply only when the policy improvement operator is Lipschitz-continuous, e.g., softmax, which ensures the limiting ODE is 'very smooth.' Our DI framework is rich enough to handle even discontinuous cases, which is true, e.g., with $\epsilon$-greedy policies.

Finally, we remark that a few complicated variants of approximate Q-learning have already been analyzed using DI based approaches (Maei et al., 2010; Bhatnagar & Lakshmanan, 2016; Avrachenkov et al., 2021). However, the need for using a DI there is fundamentally different to the one in ours (the use of $\epsilon$-greedy exploration). Maei et al. (2010) and Avrachenkov et al. (2021) look at variants of approximate Q-learning with a *fixed behavior policy* that represent *full gradient descent* with respect to suitably defined objective functions. A DI arises there due to discontinuities in the sub-gradient of the max of Q-function estimates. In contrast, Bhatnagar & Lakshmanan (2016) propose a two-timescale variant of approximate Q-learning where the max is replaced by a gradient search in the space of *smoothly continuous policies*. A DI is natural there since the gradient search can lead to one of multiple local equilibria. These differences make their DIs very different to the ones we study; hence, these analyses do not carry over to the case with discontinuous behavior policy changes.

---

[2]Linear Q-learning (resp. linear SARSA) is Q-learning (resp. SARSA) with linear function approximation.

[3]Recall the main purpose for using replay buffers was to break correlations between samples.

## 2 OUR FRAMEWORK AND MAIN RESULT

Take some function $f : \mathbb{R}^d \to \mathbb{R}^d$ and consider the generic update rule

$$\theta_{n+1} = \theta_n + \alpha_n[f(\theta_n) + M_{n+1}], \qquad n \geq 0. \tag{2}$$

The main steps of our framework for analyzing such an update rule are as follows.

1. *Partition $\mathbb{R}^d$ into regions $(\mathcal{R}_i)$ over which $f$ is 'simple'*: The word simple is subjective and will depend on the algorithm and the tools available for analysis. For approximate value-based RL methods with $\epsilon$-greedy exploration, we consider regions where the $\epsilon$-greedy policy is constant. Under linear function approximation, $f$ restricted to these regions turns out to be *linear* and continuous. However, $f$ changes discontinuously when we move from one region to the other.

2. *Use the 'Filippov' idea to stitch the different $f$-pieces and get a DI*: Formally, we combine the different pieces via the set-valued map $h : \mathbb{R}^d \to 2^{\mathbb{R}^d}$ given by

$$h(\theta) = \bigcap_{\delta > 0} \overline{\text{co}}(f(B(\theta, \delta))), \tag{3}$$

where $\overline{\text{co}}$ is the convex closure, and $B(\theta, \delta)$ and $f(B(\theta, \delta))$ denote the open ball of radius $\delta$ centered at $\theta$, and its image under $f$, respectively. An intuitive explanation for $h(\theta)$ is that it is a single update direction $\{f(\theta)\}$ within a region of the partition, and the set of all possible update directions in the vicinity of $\theta$ at the boundary between two regions. This construction is standard in control theory to handle discontinuous dynamics (Filippov, 2013) and ensures continuity of $h$ in a set-valued sense. The (deterministic) DI that captures the overall dynamics is

$$\dot{\theta}(t) \in h(\theta(t)). \tag{4}$$

Note that, unlike a differential equation, a DI uses $\in$ instead of $=$. Further, the RHS now can potentially be a set of several $\mathbb{R}^d$-valued vectors instead of a singleton. A solution of a DI is any map $\theta : [0, \infty) \to \mathbb{R}^d$ that satisfies (4). However, unlike in ODEs, solutions of a DI need not be unique. In fact, this is one of the primary sources for the pathological behaviors in linear Q-learning and SARSA(0) with $\epsilon$-greedy exploration.

3. *Use the DI to explain the limiting dynamics of* (2): Theorem A.1, taken from (Borkar, 2009; Ramaswamy & Bhatnagar, 2017), gives a set of conditions under which (2) asymptotically tracks the solution trajectories of (4). By leveraging this result, the limiting behavior of (2) is explained.

We now demonstrate the usage of our framework for explaining the limiting behaviors of linear Q-learning and SARSA(0). We begin by describing our setup and these algorithms' update rules.

Let $\Delta(U)$ be the set of probability measures on a set $U$. At the heart of RL, we have an MDP $(\mathcal{S}, \mathcal{A}, \gamma, \mathbb{P}, r)$, where $\mathcal{S}$ denotes a finite state space, $\mathcal{A}$ is a finite action space equipped with a total order, $\gamma$ is the discount factor, and $\mathbb{P} : \mathcal{S} \times \mathcal{A} \to \Delta(\mathcal{S})$ and $r : \mathcal{S} \times \mathcal{A} \times \mathcal{S} \to \mathbb{R}$ are deterministic functions such that $\mathbb{P}(s, a)(s') \equiv \mathbb{P}(s'|s, a)$ specifies the probability of moving from state $s$ to $s'$ under action $a$, while $r(s, a, s')$ is the one-step reward obtained in this transition. The main goal in RL is to find $Q_* \in \mathbb{R}^{|\mathcal{S}||\mathcal{A}|}$, the optimal Q-value function, associated with this MDP.

In practice, however, it is often the case that the state and action spaces are large. In such situations, to reduce the search space dimension, it is common to try and find an *approximation* to $Q_*$, instead of $Q_*$ itself. In this work, we focus on linear function approximation. That is, we presume we are given a feature matrix $\Phi \in \mathbb{R}^{|\mathcal{S}||\mathcal{A}| \times d}$ and the goal is to find a $\theta_* \in \mathbb{R}^d$ such that $Q_* \approx \Phi\theta_*$.

Two algorithms to find such a $\theta_*$ are linear Q-learning and SARSA(0) with $\epsilon$-greedy exploration. To enable a joint investigation, we build a unified learning scheme that includes the above two methods as special cases. Let $\phi^T(s, a)$, where $^T$ implies transpose, denote the $(s, a)$-th row of $\Phi$. Further, assume that $\Phi$ and $r$ satisfy the following condition.

$\mathcal{B}_1$. $\Phi$ has full column rank. Further, there exist constants $K_r, K_\phi \geq 0$ such that $\|\phi(s, a)\| \leq K_\phi$ and $|r(s, a, s')| \leq K_r$ for all $s, s' \in \mathcal{S}$ and $a \in \mathcal{A}$.

Next, let $\epsilon \in (0, 1]$ and $\pi_n^\epsilon : \mathcal{S} \to \Delta(\mathcal{A})$ the $\epsilon$-greedy policy at time $n \geq 0$. That is, if $\theta_n$ is the estimate of our unified algorithm at time $n$, then let

$$\pi_n^\epsilon(a|s) = \begin{cases} 1 - \epsilon(1 - 1/|\mathcal{A}|) & \text{if } a = \arg\max_{a'} \phi^T(s, a')\theta_n, \\ \epsilon/|\mathcal{A}| & \text{otherwise.} \end{cases} \tag{5}$$

We presume that $\arg\max$ breaks ties between actions that have the same value of $\phi^T(s,a)\theta_n$ using the total order on $\mathcal{A}$. Since the state and action spaces are finite, the number of $\epsilon$-greedy policies is also finite. Suppose these policies satisfy the following condition.

$\mathcal{B}_2$. The Markov chains induced by each of the finitely many $\epsilon$-greedy policies are ergodic and, hence, have their own unique stationary distributions.

Finally, let $\epsilon' \in [0,1]$. Then, given some initial estimate $\theta_0 \in \mathbb{R}^d$, the update rule of our unified algorithm at time $n \geq 0$ is

$$\theta_{n+1} = \theta_n + \alpha_n \delta_n \phi(s_n, a_n)$$
$$\delta_n = r(s_n, a_n, s_n') + \gamma \phi^T(s_n', a_n')\theta_n - \phi^T(s_n, a_n)\theta_n, \tag{6}$$

where $\alpha_n$ is the stepsize, $s_n$ is the current state at time $n$ which we presume is independently sampled from the stationary distribution corresponding to the Markov chain[4] induced by $\pi_n^\epsilon$, $a_n \sim \pi_n^\epsilon(\cdot|s_n)$, while $s_n' \sim \mathbb{P}(\cdot|s_n, a_n)$ and $a_n' \sim \pi_n^{\epsilon'}(\cdot|s_n')$. Clearly, (6) with $\epsilon' = 0$ and $\epsilon' = \epsilon$ corresponds to linear Q-learning and SARSA(0) with $\epsilon$-greedy exploration, respectively. In particular, the $\max$ operator for Q-learning is implicitly specified via the manner in which action $a_n'$ is sampled (from an $\epsilon'$-greedy policy). Note that by independently sampling the state $s_n$ in (6) from $\pi_n^\epsilon$'s stationary distribution, we ensure an idealized version of experience replay from the original DQN algorithm.

Next, suppose that the stepsize sequence $(\alpha_n)_{n\geq 0}$ satisfies the standard Robbins-Monro condition:

$\mathcal{B}_3$. $\sum_{n\geq 0} \alpha_n = \infty$ and $\sum_{n\geq 0} \alpha_n^2 < \infty$.

We are ready to analyze (6). For $n \geq 0$, let $\mathcal{F}_n := \sigma(\theta_0, s_0, a_0, s_0', \ldots, s_{n-1}, a_{n-1}, s_{n-1}', a_{n-1}')$, $f(\theta_n) := \mathbb{E}[\delta_n \phi(s_n, a_n)|\mathcal{F}_n]$, and $M_{n+1} := \delta_n \phi(s_n, a_n) - f(\theta_n)$. Then, it is easy to see that (6) can be rewritten in the form given in (2).

As per Step 1 in our framework, we partition $\mathbb{R}^d$ such that the function $f$ above has a simple form in each region of this partition. For any deterministic policy $\mathbf{a} \equiv (\mathbf{a}(s))_s \in \mathcal{A}^\mathcal{S}$, let $\mathcal{P}_\mathbf{a} := \{\theta \in \mathbb{R}^d : \forall s \in \mathcal{S}, \mathbf{a}(s) = \arg\max_a \phi^T(s,a)\theta\}$, where $\arg\max$ breaks ties using the total order. Clearly, at each $\theta \in \mathcal{P}_\mathbf{a}$, $\mathbf{a}$ is the greedy policy corresponding to the Q-value function $\Phi\theta$. Alternatively, $\mathcal{P}_\mathbf{a}$ is the *greedy region* associated to $\mathbf{a}$. For some $\mathbf{a}$, it is possible that $\mathcal{P}_\mathbf{a} = \emptyset$. Nonetheless, $\{\mathcal{P}_\mathbf{a} : \mathbf{a} \in \mathcal{A}^\mathcal{S}\}$ partitions $\mathbb{R}^d$, i.e., for any $\theta \in \mathbb{R}^d$, there is a unique $\mathbf{a}$ such that $\theta \in \mathcal{P}_\mathbf{a}$.

Let $\pi_\mathbf{a}^\epsilon$ (resp. $\pi_\mathbf{a}^{\epsilon'}$) be the $\epsilon$-greedy (resp. $\epsilon'$-greedy) policy associated with the policy $\mathbf{a}$. It is easy to see that $\pi_n^\epsilon = \pi_\mathbf{a}^\epsilon$ and $\pi_n^{\epsilon'} = \pi_\mathbf{a}^{\epsilon'}$ whenever $\theta_n \in \mathcal{P}_\mathbf{a}$. Next, let $d_\mathbf{a}^\epsilon$ denote the stationary distribution associated with the Markov chain induced by $\pi_\mathbf{a}^\epsilon$. Then, when $\theta_n \in \mathcal{P}_\mathbf{a}$, the expectations of the different terms in $\delta_n \phi(s_n)$ conditional on $\mathcal{F}_n$ are

$$b_\mathbf{a} = \mathbb{E}[\phi(s_n, a_n)r(s_n, a_n, s_n')] = \Phi^T D_\mathbf{a}^\epsilon \mathbf{r} \tag{7}$$

and

$$A_\mathbf{a} = \mathbb{E}[\phi(s_n, a_n)\phi^T(s_n, a_n) - \gamma\phi(s_n, a_n)\phi^T(s_n', a_n')] = \Phi^T D_\mathbf{a}^\epsilon(\mathbb{I} - \gamma P_\mathbf{a}^{\epsilon'})\Phi. \tag{8}$$

Here, $D_\mathbf{a}^\epsilon$ is the diagonal matrix of size $|\mathcal{S}||\mathcal{A}| \times |\mathcal{S}||\mathcal{A}|$ whose $(s,a)$-th diagonal entry is $d_\mathbf{a}^\epsilon(s)\pi_\mathbf{a}^\epsilon(a|s)$, $\mathbf{r}$ is the $|\mathcal{S}||\mathcal{A}|$-dimensional vector whose $(s,a)$-th coordinate is $\mathbf{r}(s,a) = \sum_{s'\in\mathcal{S}} \mathbb{P}(s'|s,a)r(s,a,s')$, while $P_\mathbf{a}^{\epsilon'}$ is the matrix of size $|\mathcal{S}||\mathcal{A}| \times |\mathcal{S}||\mathcal{A}|$ such that $P_\mathbf{a}^{\epsilon'}((s,a),(s',a')) = \mathbb{P}(s'|s,a)\pi_\mathbf{a}^{\epsilon'}(a'|s')$. Note that, when $\epsilon = \epsilon'$, $A_\mathbf{a}$ and $b_\mathbf{a}$ are exactly the quantities that govern the behavior of TD(0) with linear function approximation for evaluating the policy $\pi_\mathbf{a}^\epsilon$ (e.g., (Sutton & Barto, 2018, (9.11))). The $f$ function can now be explicitly written as

$$f(\theta) = \sum_{\mathbf{a}\in\mathcal{A}^\mathcal{S}} (b_\mathbf{a} - A_\mathbf{a}\theta)\mathbb{1}[\theta \in \mathcal{P}_\mathbf{a}]. \tag{9}$$

**Remark 2.1.** *Note that $f$ is discontinuous due to the presence of the indicator function $\mathbb{1}$. However, in each $\mathcal{P}_\mathbf{a}$, it is the simple linear function $b_\mathbf{a} - A_\mathbf{a}\theta$; in particular, notice that there is no 'max.'*

Assume that the matrices $A_\mathbf{a}, \mathbf{a} \in \mathcal{A}^\mathcal{S}$, satisfy the following condition (cf. Remark 2.5).

---

[4]At any time $t \geq 0$, this Markov chain moves from state $s$ to $s'$ with probability $\sum_a \pi_n^\epsilon(a|s)\mathbb{P}(s'|s,a)$.

$\mathcal{B}_4$. For all $\mathbf{a} \in \mathcal{A}^{\mathcal{S}}$, $A_{\mathbf{a}}$ is positive definite, i.e., $\theta^T A_{\mathbf{a}} \theta > 0 \, \forall \theta \in \mathbb{R}^d$ ($A_{\mathbf{a}}$ need not be symmetric).

The DI from Step 2, i.e., (4), can now be used to understand the asymptotic behavior of (6). In relation to (4), we will say a set $\Gamma \subset \mathbb{R}^d$ is *invariant* if, for every $x \in \Gamma$, there is *some* solution trajectory $(\theta(t))_{t \in (-\infty, \infty)}$ of (4) with $\theta(0) = x$ that lies entirely in $\Gamma$. An invariant set $\Gamma$ is additionally said to be *internally chain transitive* if it is compact and, for $x, y \in \Gamma$, $\epsilon > 0$, and $T > 0$, there exist $m \geq 1$ and points $x_0 = x, x_1, \ldots, x_{m-1}, x_m = y$ in $\Gamma$ such that one of the solution trajectories of (4) initiated at $x_i$ meets the $\epsilon$-neighborhood of $x_{i+1}$ for $0 \leq i < m$ after a time that is equal or larger than $T$. Such characterizations are useful to restrict the possible sets to which (6) could converge to.

Step 3 of our framework leads to our main result.

**Theorem 2.2** (**Main Theorem**: SRI-DI connection). *Suppose $\mathcal{B}_1, \ldots, \mathcal{B}_4$ hold. Then, almost surely, the iterates of (6) are stable, i.e., $\sup_n \|\theta_n\| < \infty$, and converge to a (possibly sample path dependent) closed connected internally chain transitive invariant set of the DI in (4).*

*Proof Outline.* The crucial element in the proof is Theorem A.1, originally proved in (Borkar, 2009; Ramaswamy & Bhatnagar, 2017), that discusses the stability and convergence of discontinuous update rules such as (2) and their connections to DIs. The details are in Appendix A. □

**Remark 2.3.** *The almost sure convergence in Theorem 2.2 shows that the limiting DI in (4) captures all possible asymptotic behaviors of our unified scheme given in (6).*

**Remark 2.4.** *If the only internally chain transitive invariant sets for the DI in (4) are isolated equilibrium points, then $(\theta_n)$ almost surely converges to a (possibly sample path dependent) equilibrium point. In that case, our generic scheme given in (6) will not exhibit parameter chattering.*

**Remark 2.5.** *Assumption $\mathcal{B}_4$ is for ensuring stability, i.e., a.s. boundedness of the sequence $(\theta_n)$. For linear SARSA(0) with $\epsilon$-greedy exploration, $\mathcal{B}_4$ follows from $\mathcal{B}_2$ (Gordon, 2000). For Q-learning, $\mathcal{B}_4$ need not hold in general. In that case, Theorem 2.2's claim holds a.s. on the event where the iterates are stable: $\{\sup_{n \geq 0} \|\theta_n\| < \infty\}$; see the first extension in Section 2.2. of (Borkar, 2009).*

## 3 ILLUSTRATION VIA NUMERICAL EXAMPLES

We illustrate here how Theorem 2.2 can be used to explain the limiting dynamics of linear Q-learning and SARSA(0) algorithms with $\epsilon$-greedy exploration. Specifically, we consider a range of 2-state 2-action MDPs, whose rewards, probability transition matrices and state-action features have been generated via random search (each MDP's details, such as probability transition function, reward function, discount factor, and the algorithm's exploration parameter $\epsilon$, are provided in the Appendix). In each case, the MDP and features induce 2 greedy policies, for simplicity. Even in these 'simple' settings, we see that the algorithms' trajectories exhibit complex and non-trivial phenomena. We then explain their causes via analysis of the associated DIs.

**A. Two self-consistent greedy regions – Multiple attractors**. The parameter space $\mathbb{R}^2$ is partitioned into 2 greedy regions (see definition of $\mathcal{P}_{\mathbf{a}}$ in Section 2) $\mathcal{P}_{\mathbf{a}_1}$ and $\mathcal{P}_{\mathbf{a}_2}$ via greedy policies $\mathbf{a}_1$ and $\mathbf{a}_2$, respectively.

Let us call the point $A_{\mathbf{a}_i}^{-1} b_{\mathbf{a}_i}$ a *landmark* for the partition $\mathcal{P}_{\mathbf{a}_i}$, $i = 1, 2$, where the $A_{\mathbf{a}}$ and $b_{\mathbf{a}}$ are as in (7),(8) (all matrices our examples will be positive-definite, hence invertible). The landmark $A_{\mathbf{a}_i}^{-1} b_{\mathbf{a}_i}$ is the (unique) point to which the ODE $\dot{\theta}(t) = b_{\mathbf{a}_i} - A_{\mathbf{a}_i} \theta$ would converge. Put another way, it is the point to which the Q-learning algorithm would converge were it to always sample actions $a_n$ in (6) according to the $\epsilon$-greedy policy based on $\mathbf{a}_i$.

The greedy regions in this example are both *self-consistent*. By this, we mean that each contains its own landmark: $A_{\mathbf{a}_i}^{-1} b_{\mathbf{a}_i} \in \mathcal{P}_{\mathbf{a}_i}$ for $i = 1, 2$. The landmarks, along with their respective color-shaded greedy regions, are shown as diamonds in Fig. 2a. Note that since each greedy region must be a cone and there are only 2 regions, each must be a halfspace with the origin $0$ on its boundary.

Fig. 2a shows two sample paths of the iterates $\theta_n$ of Q-learning (update (6) and $\epsilon' = 0$), with $\epsilon = 0.1$ greedy exploration, $\gamma = 0.75$ discount factor, decaying stepsizes $\alpha_n = \Theta(1/n)$ and an iteration count of $20,000$, against the backdrop of the partition diagram. The starting positions, marked with black dots, were chosen arbitrarily. Note that segments of a trajectory are often parallel to each other, because the number of directions to move from an iterate is finite in a finite MDP.

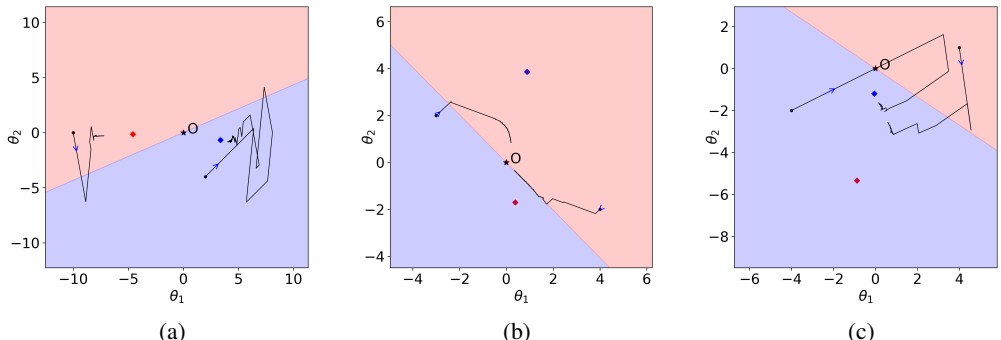

(a)       (b)       (c)

Figure 2: Trajectories under Q-learning. (Left) An MDP where function approximation in $\mathbb{R}^2$ gives rise to 2 greedy regions, both of which are self-consistent. The shaded areas represent the greedy regions, with the (self-consistent) landmarks for each partition's dynamics marked with diamonds of the respective colors. Two sample trajectories of Q-learning, with a $\Theta(1/n)$ stepsize schedule, are overlaid on the partition diagram with their starting iterates marked with black dots. The trajectories converge to any one of the self-consistent landmarks (behavior). The differential inclusion here has only the self-consistent landmarks as its asymptotically stable attractors (cause). (Center) An MDP where function approximation in $\mathbb{R}^2$ gives rise to 2 greedy regions, neither of which is self-consistent. Two sample trajectories of the iterates of Q-learning, with a $\Theta(1/n)$ stepsize schedule, are overlaid on the partition diagram with their starting iterates marked with black dots. The iterates converge to a unique point on the partition boundary (behavior) The differential inclusion here has a single 'sliding mode' attractor on the boundary (cause). (Right) An MDP where function approximation in $\mathbb{R}^2$ gives rise to 2 greedy regions, exactly one of which which is self-consistent. Two sample trajectories of the iterates of Q-learning, with a $\Theta(1/n)$ stepsize schedule, are overlaid on the partition diagram with their starting iterates marked with black dots. The iterates always converge to the unique self-consistent point in one of the partitions (behavior). The differential inclusion here has a unique attractor (cause).

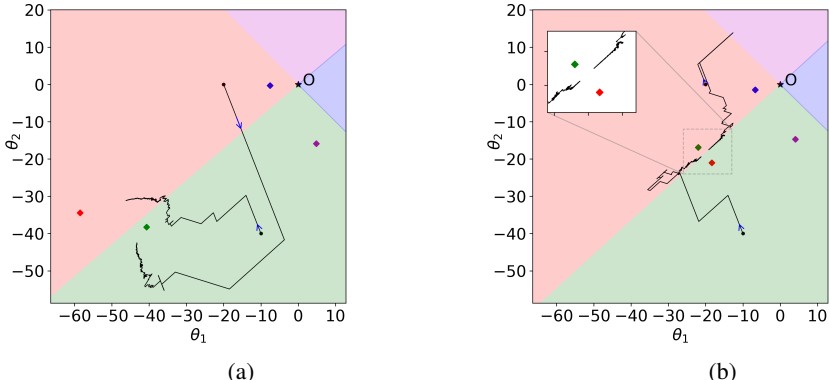

(a)                     (b)

Figure 3: Sample trajectories of the iterates of Q-learning and SARSA(0), with a $\Theta(1/n)$ stepsize schedule, overlaid on their greedy regions. (Left) In the landmark structure induced by Q-learning, there is no 'sliding mode' point on the boundary of the green and red partitions, thus the trajectories move towards the red landmark in this case. (Right) In the landmark structure induced by SARSA(0), there is a 'sliding mode' equilibrium point on the boundary of the green and red regions towards which both trajectories move (see inset).

Though the trajectories may traverse both greedy regions, they eventually approach one of the two landmarks (this observation remains true after several repeated trials). How can this be explained via our theoretical framework?

To answer this, let us look at the DI $\dot{\theta}(t) \in h(\theta(t))$ induced by Q-learning for this MDP, obtained via (9). Specifically, $h(\theta)$ equals the singleton set $\{b_{\mathbf{a}_1} - A_{\mathbf{a}_1}\theta\}$ for $\theta$ lying in the interior of the blue region (say $\mathcal{P}_{\mathbf{a}_1}$), the singleton set $\{b_{\mathbf{a}_2} - A_{\mathbf{a}_2}\theta\}$ in the interior of the red region (say $\mathcal{P}_{\mathbf{a}_2}$), and the

connected segment co $(\{b_{\mathbf{a}_1} - A_{\mathbf{a}_1}\theta, b_{\mathbf{a}_2} - A_{\mathbf{a}_2}\theta\})$ whenever $\theta$ is on the (1-dimensional) boundary of the two regions. In this DI, each point in the interior of a greedy region suffers a drift 'towards' the region's landmark. Points on the boundary, however, can be subjected to *any* drift in the convex hull of the drifts induced by each partition's affine dynamics at that point.

Theorem 2.2 says that with probability 1, the iterates must converge to an invariant and internally chain transitive set of the DI. Recall that an invariant set $I$ means that $\forall \theta \in I$ there is some trajectory $\tilde{\theta}(t)$, $-\infty < t < \infty$ lying *entirely* in $I$ and passing through $\theta$ at time 0. Roughly, for $I$ to be internally chain transitive, for any pair of points $\theta_{\text{start}}, \theta_{\text{end}} \in I$, a closeness tolerance $\eta$ and a travel time $T$, we must be able to find a solution of the DI and intermediate waypoints in $I$ such that the solution starts close to $\theta_{\text{start}}$, ends close to $\theta_{\text{end}}$, and reaches close to each successive waypoint after time $T$.

Examining this DI leads to the conclusion that it possesses only 3 invariant, internally chain transitive *singleton* sets – the 2 limiting dynamics points, plus a point $\theta^\circ$ on the boundary whose convex hull contains the 0 vector[5] – for the following reasons. These 3 singletons are certainly invariant and internally chain transitive sets, because for each there is a solution of the DI that always stays at that point for all $-\infty < t < \infty$ by virtue of zero drift. It also follows that there can be no other invariant set for the DI, because (a) any solution of the DI that, at any time, is in the interior of a region continues within that region towards its respective landmark, and (b) any solution that, at any time, is on the boundary but not at $\theta^\circ$ must suffer a drift that eventually pushes it into some greedy region's interior or along the boundary to $\theta^\circ$. We do not give a formal proof that these are the only invariant, internally chain transitive sets of the DI; the reader is directed to standard references on DIs Aubin & Cellina (2012)

Thus, according to our result, the stochastic iterates of Q-learning must eventually approach one of these 3 isolated points. We additionally remark that the point $\theta^\circ$ on the boundary is an unstable equilibrium point, because any neighborhood around it contains points (in the interiors of $\mathcal{P}_{\mathbf{a}_1}$ and $\mathcal{P}_{\mathbf{a}_2}$) from where the DI's solutions will escape towards the landmarks. If the martingale difference noise $(M_n)$ is sufficiently 'rich', i.e., it does not put all its probability mass in a subspace of $\mathbb{R}^2$, then random chance will cause the iterates to eventually escape this unstable equilibrium point. This condition will hold if all state-actions' features span $\mathbb{R}^2$, which they indeed do.

Our analysis also brings out the fact that asymptotically (as $n \to \infty$), there are no policy oscillations (iterates jumping regions infinitely often) nor parameter oscillations (iterates that 'bounce between' two regions at a positive distance) that can occur, because the iterates must eventually stabilize around one of the two landmarks in the interior of the greedy regions.

**B. No self-consistent greedy regions – Sliding mode attractor**. We consider an MDP and feature set that induce 2 greedy regions, each of which has its landmark in the *other* region. Trajectories of Q-learning, with the same parameters and stepsizes as before, appear to move eventually towards the boundary between the two regions (Fig. 2b).

Reasoning about the DI in this case, and its solutions, paints a qualitatively different picture than that of the previous example. As before, for each point $\theta$ in the interior of any greedy region, there is a singleton, nonzero drift $h(\theta) = \{b - A\theta\}$, based on the $b, A$ of the region containing $\theta$. However, this drift is directed toward the *other* region. It follows that any trajectory of the DI initialized at a region's interior point $\theta$ will reach the boundary in finite time, which rules out interior points from belonging to any internally chain transitive sets.

The only possible invariant, internally chain transitive sets, then, are to be found on the 1-dimensional boundary between the 2 regions. A moment's thought convinces us that the only such set is the singleton comprising the unique point $\theta^\star$ on the boundary for which both dynamics' drifts point in opposite directions (equivalently, 0 is in their convex hull). This point is strikingly different than the point $\theta^\circ$ of the previous example in that is a globally *stable* equilibrium point – indeed, any deviation from this point pushes the DI's trajectory *towards* it rather than *away from* it. Moreover, observe that any trajectory of the DI started from an interior point of a region first reaches the boundary in finite time, due to a positive 'velocity' towards the boundary. Thereafter, it 'slides'

---

[5]The coordinates of this (2-dimensional) point can be analytically obtained by solving 2 equations expressing that: (1) $\theta^\circ$ lies on the boundary of the greedy regions, (2) the vectors $b_{\mathbf{a}_{\text{red}}} - A_{\mathbf{a}_{\text{red}}}\theta^\circ$ and $b_{\mathbf{a}_{\text{blue}}} - A_{\mathbf{a}_{\text{blue}}}\theta^\circ$ point in opposite directions.

along the boundary to reach $\theta^\star$, i.e., $\theta(t)$ always belongs to the boundary $\forall t \geq t_0$ for some time $t_0$. Such a trajectory is termed a *sliding mode* and has been extensively studied in control theory (see e.g., Utkin (2013)).

To our knowledge, the existence of sliding modes and associated equilibria has not been previously demonstrated for approximate RL algorithms. Also, our main result (Theorem 2.2) guarantees that asymptotically, the stochastic iterates $\theta_n$ will not oscillate or chatter, although policy oscillations are possible (the iterates may cross the boundary infinitely often on their way to $\theta^\star$).

**C. Only one self-consistent greedy region – Unique landmark attractor**. We consider an MDP that, with linear Q-function approximation in $\mathbb{R}^2$, induces two greedy regions with exactly *one* of them being self-consistent (Fig. 2c). The DI for this setting has the property that points in the red region will experience a (vector) drift, towards the boundary, which is bounded away from $0$ in norm. As a result, any DI trajectory that starts in the red region will, in finite time, hit the boundary. The driving (set-valued) function $h(\theta)$ for each boundary point can be shown to have all its constituent drift vectors 'pointing into' the blue region, so any solution from the boundary must instantaneously leave it. Continuing from here, the trajectory moves towards the blue landmark, which can be shown to be the sole invariant, internally chain transitive set of the DI. It is thus clear, from Theorem 2.2, that all the original stochastic iterates of the algorithm must eventually approach this point. This is a situation in which no policy or parameter chattering occurs. Also, as observed by Young and Sutton Young & Sutton (2020), it is possible that this (single) landmark point to which every trajectory converges may represent a very poor quality approximation for the optimal value function.

Figs. 3a and 3b show that even with the same MDP and features (and hence greedy regions), Q-Learning and SARSA(0) can induce different landmark structures, resulting in different trajectory behavior. We discuss other examples such as the chattering phenomenon observed in Gordon (1996); Young & Sutton (2020) in the Appendix.

# 4   CONCLUSION AND FUTURE DIRECTIONS

We introduced the novel toolset of SRIs and DIs to help understand the algorithmic dynamics of approximate RL algorithms. We may have merely scratched the surface with regard to wielding the DI approach to its fullest potential. Our framework paves the way for a more comprehensive understanding of existing approximate-RL algorithms as well as the design of new schemes with 'well-behaved' DIs, hopefully resulting in value beyond just analysis to the realm of synthesis.

On a somber note, the insights we have uncovered about algorithms using arguably the simplest possible (linear) function approximation casts doubt on their utility in more complicated, nonlinear approximation architectures. It is plausible that there are many more failure modes in these settings than the range observed in the linear case, as also seen by Young & Sutton (2020) in neural approximation. Even with linear function approximation, we have shown in this paper that (a) one could converge to multiple attractors with disparate qualities, (b) one could oscillate between multiple policies' greedy regions, making it difficult to devise simple stopping criteria (c) even if one converges always to a unique point within a greedy partition, the induced greedy policy could be arbitrarily bad – these spell doom for the practitioner unless carefully addressed and investigated. Separately, we acknowledge that our theory addresses only linear function approximation and more work is needed to extend it to nonlinear approximation architectures.

Yet another promising direction for future research, which is not taken up in this paper, is that of the quality of the limiting iterates (assuming there is some reasonable form of convergence). Again, Young & Sutton (2020, 'Worst-case example') and Bertsekas (2011) point out that approximate policy iteration schemes could very well converge to the 'worst possible outcome', which, perhaps, with suitable design considerations based on the DI connection, can be avoided in new algorithms to come. For instance, a whole new class of algorithms can arise by assigning arbitrary values of $\epsilon, \epsilon'$ in the generic update (6) with favourable properties. On a slightly different but related note, our work also reinforces the fact that merely ensuring stability of an incremental RL algorithm's iterates is by no means sufficient to guarantee good performance – discontinuous policy updates often swamp out gains from stability by inducing complex and varied convergence modes.

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
