# OpenReview forum: "Demystifying Approximate RL with $\epsilon$-greedy Exploration: A Differential Inclusion View"
_ICLR.cc/2023/Conference — Submitted to ICLR 2023_

### Official Review · Reviewer_svEP · 2022-10-22

**Confidence:** 2
**Correctness:** 4
**Technical Novelty And Significance:** 4
**Empirical Novelty And Significance:** 4
**Recommendation:** 8

**Clarity, Quality, Novelty And Reproducibility:**

The paper is generally clear and easy to follow. The explanations of the concepts are detailed and thorough.

The results presented in the paper seem novel.

Given the information presented in the paper, I suspect that the results would be reproducible.

One small suggestion would be to replace the phrase “Q-value function” with “action-value function” everywhere it is used. (This is not affecting my rating, it is only my personal preference.)


**Strength And Weaknesses:**

The main strength of this paper is its comprehensive introduction of the new analysis framework that provides previously missing theory on how to explain unexpected behaviors in certain algorithms under approximation.

**Summary Of The Paper:**

This paper presents a new analysis framework based on differential inclusion (DI) theory for analyzing value-based reinforcement learning (RL) methods with discontinuous behavior policy changes (such as Q-learning with an epsilon-greedy behavior.) Using the new framework, the paper provides an explanation of the asymptotic behaviors of both Sarsa(0) and Q-learning with an epsilon-greedy policy and linear function approximation. Finally, the paper discusses several MDP examples and shows how an algorithm’s limiting DI affects its convergence behavior.

**Summary Of The Review:**

This paper presents a new framework to analyze the convergence behaviors of algorithms with discontinuous behavior policies and function approximation. This work is likely of interest to those doing RL theory research, and could be of some interest to the larger RL research community as well. Therefore I vote to accept.

---

> ### Author Response · Authors · 2022-11-19
> **Response to Reviewer svEP**
>
> Thank you for your encouraging feedback. We will seriously consider your suggestion to use the phrase ``action-value function" instead of "Q-value function."

---

### Official Review · Reviewer_ieqE · 2022-10-24

**Confidence:** 3
**Correctness:** 3
**Technical Novelty And Significance:** 2
**Empirical Novelty And Significance:** 2
**Recommendation:** 5

**Clarity, Quality, Novelty And Reproducibility:**

=== Clarity ===

The paper is overall easy to follow, but presentation-wise it can be improved by better motivating the problem to be studied.

=== Quality ===

The paper seems to have solid theoretical results, though it is not clear to me how significant are the results and how they can be useful.

=== Novelty ===

The paper is relatively novel in that it applies the new perspective of DI to understand the jittering behavior of value-based algorithms.

=== Reproduce ===

It should be straightforward to reproduce results in the paper.

**Details Of Ethics Concerns:**

No concerns.

**Strength And Weaknesses:**

=== Strength ===

The paper seems to provide a very interesting toolkit to systematically understand the behavior of value-based learning algorithm under linear FA and with epsilon-greedy exploration, which is a useful abstraction for many commonly implemented RL algorithms. This might pave the way for future investigations into similar topics.

=== Weakness ===

A potential weakness of the paper is that it is not clear what are the implications of the results in the paper and how significant they are. The main theoretical statement Thm 2.1 claims that under certain conditions, parameter iterates of such value-based algos can jitter within a set of solutions. There is little characterization of how the set of solutions relate to each other (e.g., quality of the solutions) and how one can make use of the theoretical characterization in practice.

**Summary Of The Paper:**

The paper proposes to apply theory of differential inclusion to understand and explain the behavior of epsilon-greedy based value-learning algorithms with function approximation, such as Q-learning and SARSA(0) with epsilon-greedy exploration. The main result (Thm 2.1) characterizes the asymptotic behavior of aforementioned algorithms under a few assumptions.

**Summary Of The Review:**

=== **Background question** ===

I would like to clarify a few sentences in the intro section of the paper. The authors write

"The expected behavior here would be convergence to the optimal policy..."

Is the behavior expected because the representation matrix Phi contains the optimal Q-function as one of its columns, so that we should expect the linear function approximation to be able to learn the optimal Q-function and therefore optimal policy. In other words, the "expected" here is purely based on realizability of the target Q-function in the function class.

In practice, such convergence behavior is not observed. I wonder intuitively if this can be explained by the fact that though the optimal Q-function can be represented within the class of linear functions, the algorithm itself can take a path that does not lead to the optimal Q-function. This reminds me of certain conclusions in the policy gradient case, where if the function class is not closed under the policy improvement operator, there is no guarantee to convergence even though optimal Q-function is representable. Can the authors comment on the potential connection here? For general audience, it would also be interesting to understand more of the intuitions of the jittering / non-convergence behavior.

[1] Bhandari et al, 2019, Global Optimality Guarantees For Policy Gradient Methods

=== **Textbook instability** ===

I am a bit confused by the use of "textbook instability". What does this instability refer to exactly?

=== **$h$ not Lipschitz continuous***

I find the background under Eqn 1 a bit confusing in wording. If I understand correctly, $h$ is Lipschitz continuous when doing policy evaluation, as the system is linear; when doing control with $\epsilon>0$-greedy, $h$ is generally not Lipschitz continuous. Is this the case? Maybe worth clarifying more here.

=== **Main result** ===

Assumption B4 I think need further explaining. From the background under Eqn 3, $\epsilon>0,\epsilon'=0$ corresponds to Q-learning; $\epsilon'=\epsilon>0$ corresponds to SARSA(0). The $A_a$ matrix is
$$\Phi^T D_a^\epsilon (I-\gamma P_a^{\epsilon'}) \Phi$$
I think it is true that when $\epsilon'=\epsilon$, the above matrix is PD i.e., $x^T A_a x>0$ for any vector $x$, because essentially $D_a^\epsilon$ and $P_a^{\epsilon'}$ are diagonal matrix and one-step transition matrix of the same $\epsilon$-greedy policy.

However, it is not immediately clear to me why in the Q-learning case $\epsilon'=0,\epsilon>0$ the $A_a$ matrix is also PD. If B4 is assumed true, it seems to be a very big assumption? I wonder having assumption B4 in place would be in conflict with much of the prior literature in analyzing Q-learning with FA, where it seems that obtaining theoretically justified convergence would need to impose stringent assumption on the data process, such as near on-policy [1].

[1] Melo et al, 2008, An Analysis of Reinforcement Learning with Function Approximation

=== **Presentation** ===

Presentation-wise, I'd certainly appreciate more comprehensive background introduction on DI. The whole paper is pretty dense, especially in Sec 2, where it seems tricky for general audience to understand the main technical points of the paper unless equipped with sufficient background in DI.

---

> ### Author Response · Authors · 2022-11-19
> **Reponse to Reviewer ieqE - Part 1**
>
> 1. **Implications and significance of the result**: Our work does not claim to explain *only* policy oscillation (this is what we presume the reviewer means by "jittering"). Instead, our main result is that *all* asymptotic behaviors (including policy oscillation, and convergence to unique/multiple attractors on different runs) can be explained in terms of the limiting DI. In particular, the proof argument shows that the algorithm's iterates asymptotically track solutions of the limiting DI. Hence, all that is needed to understand the algorithm's limiting behavior is the nature of the limiting DI.
> ---
>
> 2. **About the nature of solutions of the DI and how they relate to each other**: Note that the connection to a DI, as opposed to a differential equation, is itself a nontrivial step. To the best of our knowledge, such a perspective has not been used earlier in the RL community to study algorithms with $\epsilon$-greedy type exploration or, more generally, algorithms whose driving functions have discontinuities  (in fact, there exist no analyses in the function approximation setting). How the nature of the limiting DI can be understood and exploited in practice is an open question. Still, we believe our work provides a new fundamental pathway even to attempt to answer such questions.
> ---
>
> 3. **About the usefulness of the results in practice**: Our work will help a practitioner systematically understand behaviors of value-based RL algorithms observed in practice. Given that a practitioner has observed some behaviors, our theory serves to illuminate their causes and thus may be a key step in root-cause analysis and further debugging. In particular, our theory provides a template of all possible behaviors that can occur in practice (via the study of the associated DI and its structure). Moreover, our work indicates to the practitioner that these behaviors are *fundamental* to the underlying setting and cannot be avoided by superficial tweaks to the algorithm, such as choosing a `better algorithm parameter.'
> ---
>
> 4. **On why we expect linear DQN to learn the optimal (realizable) Q-function in Figure 1**: First, we have significantly updated our setting to illustrate the unreliability issue in linear DQN even more clearly. Now, not only is $Q_*$ realizable, but (i) the initial behavior policy is also set to be $\epsilon$-greedy version of the optimal policy, and (ii) $\theta_0$ is also close to $\theta_*.$ In this idealized setting, linear Q-learning with a fixed behavior policy (initialized as above) is guaranteed to converge to $\theta_*$ (Melo et al. 2008). Thus, in this setting, it is `natural' to expect linear DQN to also go to $\theta_*.$
> ---
>
> 5. **Intuition for why convergence to a perfectly realizable $Q_*$ does not reliably take place in Figure 1**: Yes, you are generally correct in suggesting that ".. though the optimal Q-function can be represented within the class of linear functions, the algorithm itself can take a path that does not lead to the optimal Q-function." More specifically, by applying our framework, it turns out that the algorithm's iterates follow *simple* linear dynamics in various regions (corresponding to greedy policies) in the parameter (i.e., approximate value function) space. However, the drift of these linear dynamics can be very different in each region. As a result, the iterates get pushed in very different ways depending on the current greedy behavior policy. This can effectively prevent iterates in *faraway* regions from successfully getting drawn towards the optimal $Q_*$ by getting `locally trapped' in or on the boundary of such regions. At a high level, this resembles how policy gradient methods can get trapped near suboptimal local maxima (i.e., locally optimal policies). However, bear in mind that for the setting in our paper, the places where the iterates get trapped do not necessarily admit interpretation as local maxima of suitable value functions (in fact, in the sliding mode/policy oscillation situation, the linear drift of the iterates can be nonzero on both sides of the boundary, yet the iterates may settle on the boundary).
> ---
> 6. **On "textbook instability"**: This refers to the phenomenon in which the iterates of value-based RL algorithms with (linear) function approximation can grow without bound ($\sup_n \\| \theta_n \\| = \infty$). Standard counterexamples, e.g., by Baird and Tsitsiklis-van Roy, illustrate this effect.
> ---
> 7. **On Lipschitz continuity of the function $h$ and the explanation under eq. (1)**: The driving function $h$ in eq. (1) is indeed Lipschitz continuous when doing policy evaluation (i.e., when the behavior policy is fixed). However, when doing control with $\epsilon$-greedy sampling and $\epsilon < 1$, $h$ loses not just Lipschitz continuity (as you say) but *continuity altogether*. We have revised the explanation under eq. (1) to better reflect this property.

---

> > ### Author Response · Authors · 2022-11-19
> > **Reponse to Reviewer ieqE - Part 2**
> >
> > 8. **On Assumption $\mathcal{B}_4$**: This assumption is vacuous for linear SARSA(0) since it follows from $\mathcal{B}_2.$  For linear Q-learning, this is a minimal assumption for guaranteeing almost sure stability. However, this assumption can be dropped. In that case, along every sample path, the iterates will either diverge to infinity or remain stable. The conclusion of our main result continues to hold on the latter event. This is because our proof operates on a per-sample path basis. We have highlighted this in Remark 2.5 using the following words: Assumption $\mathcal{B}_4$ is for ensuring stability, i.e., a.s. boundedness of the sequence $(\theta_n).$  For linear SARSA(0) with $\epsilon$-greedy exploration, $\mathcal{B}_4$ follows from $\mathcal{B}_2$ (Gordon, 2000). For Q-learning, *$\mathcal{B}_4$* need not hold in general. In that case, the conclusion of Theorem  2.2 holds a.s. on the event where the iterates are stable: $\{\sup_{n \geq 0} \\|\theta_n\\| < \infty\}$; see the first extension in Section 2.2. of (Borkar, 2009).
> > ---
> > 9. **On Sec. 2 and background on DIs**: Thank you for your feedback. We have significantly revised Sec. 2 to include (i) an explanation of our core framework of identifying regions with *simple* dynamics followed by `stitching' the simple dynamics together, (ii) additional intuition behind *convexifying* driving functions to result in set-valued driving functions (the idea underlying the DI approach), (iii) an explanation of how the basic driving function ($f$) for Q-learning and/or SARSA(0) are piecewise linear but overall discontinuous. We also reiterate that Sec. 3 contains examples of DIs that result for simple 2-state MDP settings, with details about the structure of their solutions. Please also see our response to Reviewer JuQi on interpreting DI theory (https://openreview.net/forum?id=Ms1Zs8s7rg&noteId=XB18pdEP2r).

---

> > > ### Author Response · Authors · 2022-11-22
> > > **Feedback on revision and response**
> > >
> > > Dear Reviewer ieqE,
> > >
> > > We thank you for your constructive comments. Please let us know if our revision and comments helped resolve your concerns. Please also let us know if there are any new concerns. We will be happy to address them.
> > >
> > > We hope you had a chance to look at our simple and detailed example that elaborates on the exact role of DI theory in our work (https://openreview.net/forum?id=Ms1Zs8s7rg&noteId=XB18pdEP2r).
> > >
> > > Thank you.

---

### Official Review · Reviewer_A5T2 · 2022-10-27

**Confidence:** 3
**Correctness:** 3
**Technical Novelty And Significance:** 3
**Empirical Novelty And Significance:** Not applicable
**Recommendation:** 5

**Clarity, Quality, Novelty And Reproducibility:**

As aforementioned, this work examines the origin of the asymptotic behavior of two well-known approximate RL algorithms that constitutes a quite challenging task. For this purpose, authors resort to the  Differential Inclusion (DI) theory. Nevertheless, the intuition behind the usage of the differential Inclusion (DI) theory is not clear. Actually, Eqs. 6 and 7 should be presented in more detail. Apart from that, it seems that the theoretical analysis is more related to the LSTD algorithm. It would be really useful if authors could elaborate if the proposed theoretical result can be also applied in the case of the LSTD algorithm. Another point that should be also discussed by the authors is the usage for explaining the behavior of the tabular versions of Q-learning and SARSA algorithms without any kind of exploration. Finally, it is not clear how the proposed theory can be used in order to overcome or fix the limitations of the approximate Q-learning and SARSA algorithms.

Other questions:

- What do you mean by saying `almost surely` in Theorem 2.1?

**Strength And Weaknesses:**

**Strength**

- The explanation of the asymptotic behavior of two well-known approximate RL algorithms, Q-learning and SARSA, with $\epsilon$-greedy exploration is quite important and constitutes a challenging open problem.
- Differential Inclusion (DI) theory can be used for analyzing value-based RL methods with discontinuous behavior policy changes.
- Numerical examples have been used to demonstrate the usage of the proposed theoretical result.

**Weaknesses**

- The intuition behind the usage of the differential Inclusion (DI) theory should be described in a more precise way.
- The theoretical analysis seems to be more related to the LSTD. Authors should elaborate if the proposed theoretical result can be also applied in the case of the LSTD algorithm.
- The usage of the proposed theorem for verification of the convergence to the optimal Q-function of the tabular versions of Q-learning and SARSA algorithms need to be examined.

**Summary Of The Paper:**

This work tries to explain the asymptotic behaviors of two well-known approximate RL algorithms, Q-learning and SARSA, with $\epsilon$-greedy exploration. The theoretical analysis presented in this work uses Differential Inclusion (DI) theory to analyze value-based RL methods with discontinuous behavior policy changes. Numerical examples have also been used to demonstrate the usage of the proposed theoretical result for explaining the limitation of  Q-learning and SARSA algorithms with linear function approximation and $\epsilon$-greedy exploration.

**Summary Of The Review:**

The theoretical analysis presented in this paper is novel, but there are some parts of the paper that are not totally clear. The most important question that needs to be answered is how we can use the proposed theory in order to overcome or fix the unexpected of the approximate Q-learning and SARSA algorithms with $\epsilon$-greedy exploration.

---

> ### Author Response · Authors · 2022-11-19
> **Response to Reviewer A5T2**
>
> 1. **On the intuition behind the usage of the DI theory**: We have now added an explicit description of the key steps in our proposed analysis framework at the start of Section 2, pg. 4. These steps can be summarized as follows:  i.) break down the parameter space into regions where the algorithm’s dynamics are simple,  ii.) identify a DI that stitches the local dynamics together, and iii.) use this DI to explain the algorithm’s overall (possibly complex) behavior. Note that a DI is a generalization of an ODE which enables this stitching by allowing for multiple update directions at every point. Please also see our response to Reviewer JuQi on interpreting DI theory (https://openreview.net/forum?id=Ms1Zs8s7rg&noteId=XB18pdEP2r).
> ---
>
> 2. **On the perceived relation of our analysis to that of LSTD**: We assume the reviewer relates our analysis to that of LSTD because the expectation of the term $\delta_n \phi(s_n, a_n)$ in (6) in the revised pdf, conditional on $\mathcal{F}_n,$ has the form $b_n - A_n \theta_n,$ and $A_n$ and $b_n$ change with $n.$ Indeed, in LSTD as well, we have a changing $A_n$ and $b_n.$ However, in LSTD, there is only one $A$ and $b$ (of the form discussed in (7) and (8) of the revised pdf) corresponding to the fixed policy being evaluated; further,$A_n$ and $b_n$ there are their online estimates. Hence, in LSTD, $A_n$ and $b_n$ will quickly settle near $A$ and $b.$ However, this is very different from the setting in our work. First, while LSTD concerns policy evaluation, our work concerns control, i.e., we analyze algorithms that attempt to find the optimal policy. Second, the changing $A$'s and $b$'s in our work are due to the changing behavior policies. These changes may or may not settle, e.g., in policy oscillation, it does not (see Figure 1b of our revised pdf).
>
> ---
>
> 3. **On using our main result in tabular settings**: This is a very interesting question. Our analysis does apply to the tabular case, and our main result holds there as well. In particular, one can see that the dynamics there is discontinuous, exactly of the form given in (9) in our revised pdf. However, the limiting DI there can be shown to have a global Lyapunov function $\\|Q - Q_\*\\|\_\infty;$  see Theorem 2, Chapter 10 of (Borkar, 2009), and hence any solution of the DI will always converge to $Q_*.$ Unfortunately, such a global Lyapunov function does not exist for the DI in general function approximation settings; if it did, we would never see the range of pathological behaviors as in Figure 1 or Section 3.
>
> ---
>
> 4. **On the usage of the term `almost surely' in Theorem 2.1**: The term almost surely is with respect to the probability distribution of states and actions that the algorithm in (6) uses.

---

> > ### Author Response · Authors · 2022-11-22
> > **Feedback on revision and response**
> >
> > Dear Reviewer A5T2,
> >
> > We thank you for your constructive comments. Please let us know if our revision and comments helped resolve your concerns. Please also let us know if there are any new concerns. We will be happy to address them.
> >
> > We hope you had a chance to look at our simple and detailed example that elaborates on the exact role of DI theory in our work (https://openreview.net/forum?id=Ms1Zs8s7rg&noteId=XB18pdEP2r).
> >
> > Thank you.

---

### Official Review · Reviewer_JuQi · 2022-11-01

**Confidence:** 2
**Correctness:** 3
**Technical Novelty And Significance:** 3
**Empirical Novelty And Significance:** 2
**Recommendation:** 5

**Clarity, Quality, Novelty And Reproducibility:**

I think this paper is overall clearly written. However, I am not an expert in Differential Inclusion (DI) theory, so I am not able to evaluate the correctness of the results.

**Strength And Weaknesses:**

Strength:

The stability issue of Q learning and SARSA is very important. The paper uses a novel technique named Differential Inclusion (DI) theory to study this problem. This technique is not very common in the RL community, and I think it's quite interesting to introduce this technique to the community.

Weakness:

1) Although epsilon-greedy is an important method, it is known that this approach is not a very smart exploration method. Therefore, it would be useful to discuss how to extend this framework to other methods.
2) The assumption B4 in page 5 seems to be quite strong. I am not sure how to interpret this assumption.
3) The experiments only use numerical simulations. It would be better to provide empirical evidence in more realistic environments.


**Summary Of The Paper:**

This paper analyzes the stability issues in Q learning and SARSA with epsilon greedy exploration. The authors use Differential Inclusion (DI) theory to analyze the behavior of Q learning and SARSA. The authors use numerical examples to illustrate their theory.

**Summary Of The Review:**

The technique that this paper introduced is interesting, but since there are several issues in the paper and thus I don't think it qualifies as an ICLR publication.

---

> ### Author Response · Authors · 2022-11-19
> **Response to Reviewer JuQi**
>
> 1. **On the knowledge that $\epsilon$-greedy approach is not smart**: We assume this statement is being made in the context of adaptive exploration strategies such as UCB and Thompson sampling. However, note that these adaptive variants can only improve the regret *rate* towards a target (benchmark). Our goal is to understand something even more fundamental than convergence rates: where, if at all, do approximate value-based RL methods with $\epsilon$-greedy exploration converge? This has been an open question (see Problem~1 in Sutton, 1999), and our work provides the first framework to even attempt to answer such a question. Our framework paves the way for future analysis of more complicated exploration strategies.
> On a separate note, we stress that the $\epsilon$-greedy exploration strategy is widely used in practice and, hence, our current work is important in its own right; see (Dann et al. 2022), which studies $\epsilon$-greedy exploration for the same reasons. In fact, to paraphrase them, "The reason for using myopic (e.g., $\epsilon$-greedy) exploration is because it is easy to implement and works well in a range of problems, including many of practical interest (Mnih et al., 2015; Kalashnikov et al., 2018). On the other hand, it is unknown how to implement existing strategic exploration (e.g., UCB / Thompson type) approaches computationally efficiently with general function approximation. They either require solving intricate non-convex optimization problems or sampling from intricate distributions."
>
>     *(Dann et al., 2022)*: Dann C, Mansour Y, Mohri M, Sekhari A, Sridharan K. Guarantees for epsilon-greedy reinforcement learning with function approximation, ICML 2022
>
>     *(Mnih et al., 2015)*: Mnih, V., Kavukcuoglu, K., Silver, D., Rusu, A. A., Veness, J., Bellemare, M. G., Graves, A., Riedmiller, M., Fidjeland, A. K., Ostrovski, G., et al. Human-level control through deep reinforcement learning. nature, 518(7540): 529–533, 2015.
>
>     *(Kalashnikov et al., 2018)*: Kalashnikov, D., Irpan, A., Pastor, P., Ibarz, J., Herzog, A., Jang, E., Quillen, D., Holly, E., Kalakrishnan, M., Vanhoucke, V., et al. Qt-opt: Scalable deep reinforcement learning for vision-based robotic manipulation. arXiv preprint arXiv:1806.10293, 2018
> ---
> 2. **On Assumption $\mathcal{B}_4$**: This assumption is vacuous for linear SARSA(0) since it follows from $\mathcal{B}_2.$  For linear Q-learning, this is a minimal assumption for guaranteeing almost sure stability. However, this assumption can be dropped. In that case, along every sample path, the iterates will either diverge to infinity or remain stable. The conclusion of our main result continues to hold on the latter event. This is because our proof operates on a per-sample path basis. We have highlighted this in Remark 2.5 using the following words: Assumption $\mathcal{B}_4$ is for ensuring stability, i.e., a.s. boundedness of the sequence $(\theta_n).$  For linear SARSA(0) with $\epsilon$-greedy exploration, $\mathcal{B}_4$ follows from $\mathcal{B}_2$ (Gordon, 2000). For Q-learning, *$\mathcal{B}_4$* need not hold in general. In that case, the conclusion of Theorem  2.2 holds a.s. on the event where the iterates are stable: $\{\sup_{n \geq 0} \\|\theta_n\\| < \infty\}$; see the first extension in Section 2.2. of (Borkar, 2009).
> ---
> 3. **On experiments only using numerical simulations**:  The true purpose of the simulation section (Sec. 3) is to complement the theory we have proposed in the sense of exploring the diverse range of possibilities that an MDP's DI can express for the limiting dynamics of a value-based RL algorithm. In this regard, the simplicity of the MDPs in the section is in fact their strength, as they are able to showcase a wide range of pathological behaviors in terms of convergence (policy oscillations, multiple attractors, convergence to the worst policy and, for the first time, sliding-mode attractors). Instead of dismissing these rudimentary examples as `simple', the natural concern to worry about is: If all this can occur in the simplest of MDPs, then what confidence might the algorithms inspire for full-blown RL with large state/action spaces and nonlinear function approximation? On a separate note, we refer the reviewer to a recent preprint by (Schaul et al., 2022), which empirically discusses the phenomenon of policy oscillation in a more realistic setting. Our work is the first to explain the causes that could lead to the observed phenomenon.
>
>     *(Schaul et al., 2022)*: Schaul, T., Barreto, A., Quan, J. and Ostrovski, G., 2022. The Phenomenon of Policy Churn. arXiv preprint arXiv:2206.00730.

---

> > ### Comment · Reviewer_JuQi · 2022-11-20
> > **Thanks for your clarification**
> >
> > Thanks for your clarification. I am not an expert in Differential Inclusion (DI) theory so I am not very confident in my review. I think this paper will benefit a lot from a better interpretation of this theory. Therefore, I decided to keep my score.

---

> > > ### Author Response · Authors · 2022-11-20
> > > **Interpretation of DI theory**
> > >
> > > We thank you for responding to our comments. The use of DI is relatively new in the RL community. Hence, we understand your concerns. However, the (relatively short) proof of our main result in Appendix A relies entirely on standard real-analysis arguments. Hence, the validity of our main result can be established without requiring the knowledge of any DI theory whatsoever. To interpret our main result and understand its implications, yes, one would need some knowledge of DIs. To help the reader with this latter task, we promise to add the following text to the appendix in the next revision.
> > >
> > > **Ground-level introduction to DIs**:  A DI is a generalization of a differential equation. Recall that one of the ways of interpreting the (autonomous) differential equation $\dot{\theta}(t) = f(\theta(t))$ is to view it as a relation between the velocity of a particle and its position. A particle's trajectory is said to be a solution of this differential equation if its velocity at every position $\theta$ is exactly $f(\theta).$ There is a similar requirement in the differential inclusion $\dot{\theta}(t) \in h(\theta(t)).$ In that, a particle's trajectory is said to be its solution if its velocity at any position $\theta$ lies in the set $h(\theta).$
> > >
> > > An easy example to understand this generalization is the following. Suppose a particle lives on the real line, and we want that its velocity be $+1$ on the positive side and $-1$ on the negative side. This dynamics cannot be studied using a differential equation lens because of the discontinuous jump from $-1$ to $+1.$ However, we can study it using the DI $\dot{\theta}(t) \in h(\theta(t)),$ where $h(\theta) =$ {-1}, {+1}, and the interval $[-1, + 1],$ for $\theta < 0,$ $\theta > 0,$ and $\theta = 0,$ respectively. In particular, since $h(0)$ is the convex closure of the set \{-1, + 1\}, it can be shown that $h$ is Marchaud, i.e., it is continuous in a set-valued sense. Like Lipschitz continuity guarantees the existence of solutions for differential equations (for any initial point), the Marchaud property guarantees the existence of solutions for DIs. Unlike differential equations, though, the solutions of a DI are not guaranteed to be unique. In the above example, the three trajectories i.) $\theta(t) = 0$ for all $t \geq 0;$ ii.) $\theta(t) = -t$ for all $t \geq 0;$ and iii.) $\theta(t) = +t$ for all $t \geq 0$ are all valid solutions of the DI with the same initial condition $\theta(0) = 0.$
> > >
> > > Our work shows that the asymptotic behaviors of linear Q-learning and SARSA(0) with $\epsilon$-greedy exploration are exactly as above. There is a partition of the space in which they live, and within each region of this partition, these algorithms have different asymptotic dynamics (see (9) in our revised pdf). In particular, this dynamics is continuous within a region but discontinuously changes from one region to another. Therefore, one cannot use differential equations to study these behaviors; instead, one needs a DI. Our main result rigorously shows that linear Q-learning and SARSA(0) indeed track suitable solutions of the DI in (4) of our revised pdf. The summary of our work is that, by understanding all the complex behaviors that DI in (4) can show, one can explain the different behaviors that value-based RL methods with $\epsilon$-greedy exploration can exhibit.

---

> > > > ### Author Response · Authors · 2022-11-22
> > > > **Feedback on revision and response**
> > > >
> > > > Dear Reviewer JuQi,
> > > >
> > > > We once again thank you for your constructive comments. We have provided a simple and detailed example to elaborate on the exact role of DI theory. Please let us know if that helped you better understand the implications of our main result. Please also let us know if there are any other aspects of DI theory or our result that you don't understand. We will be happy to address them.
> > > >
> > > > Thank you.

---

### Author Response · Authors · 2022-11-19
**Summary of Major Changes in the Revision**

We thank the reviewers for their constructive feedback. We have revised Section 1 (Introduction) and Section 2 (Our Framework and Main Result) to address the significant concerns. A summary of the important changes is as follows.

M1)  We have updated Figure 1, pg. 2, using a new MDP, to illustrate the unreliability of the linear DQN algorithm even more clearly.

M2) We have added an explicit description of the key steps in our proposed analysis framework at the start of Section 2, pg. 4 (we have also included its summary in item 1 of our key contributions on pg. 2). Further, as part of step 2, we have explained what a DI is and discussed its role in our analysis. Lastly, we have highlighted the places where the different steps of our framework are being used during our analysis of linear Q-learning and SARSA(0) on pgs. 5 and 6.

M3) We have added a discussion about Assumption $\mathcal{B}_4$ in Remark 2.5.

---

### Decision · Program_Chairs · 2023-01-20

**Decision:**

Reject

**Justification For Why Not Higher Score:**

N/A

**Justification For Why Not Lower Score:**

N/A

**Metareview: Summary, Strengths And Weaknesses:**

This work tries to explain the asymptotic behaviors of Q-learning and SARSA with $\epsilon$-greedy exploration, and the proposed tool is the differential inclusion theory.

The strength and weakness of this paper are both very clear.

Strength: the paper has some merits since it introduces some interesting new toolkits of differential inclusion to the study of RL. It can still be a good starting point of studying the limiting behavior of the discontinuous dynamics of the algorithms.

Weakness: the proved result is very weak.

(1). For SARSA with linear approximation: convergence to bad point, oscillation, and convergence to good point, are all possible.
(2). For Q-learning with linear approximation where B4 may not be true: divergence, convergence to bad point, oscillation, and convergence to good point, are all possible. That is, everything is possible, nothing is really proved.

Moreover, since the possibility to converge (or diverge/oscillate) to all type of points, there does not seem to be a way to use the proposed analysis to help the algorithm to converge to the desired points.  (The authors justification that the result may help practitioners debugging is not convincing. After incorporating all engineering tricks to make the algorithms work, the resulting code will be far away from the clean vanilla algorithm studied here. The theory may not character the real applications.)



**Summary Of Ac-Reviewer Meeting:**

N/A